# Comparison of Microbiological Profiles of Primary Hip and Knee Peri-Prosthetic Joint Infections Treated at Specialist Centers Around the World

**DOI:** 10.3390/microorganisms13071505

**Published:** 2025-06-27

**Authors:** Emin Suha Dedeogullari, Pablo Slullitel, Isabel Horton, Bulent Atilla, Saif Salih, Paul Monk, Ahmet Mazhar Tokgozoglu, Michael Goplen, Bonita Tsang, Martin Buljubasich, Hesham Abdelbary, Simon Garceau, George Grammatopoulos

**Affiliations:** 1Hacettepe University Hospital, Hacettepe, A.Adnan Saygun Cd., 06230 Altindag, Ankara, Turkey; ededeogullari@ku.edu.tr (E.S.D.); batilla62@gmail.com (B.A.); mtokgozoglu@gmail.com (A.M.T.); 2Hospital Italiano de Buenos Aires, Tte. Gral. Juan Domingo Perón 4190, CABA, Buenos Aires 1199, Argentina; pablo.slullitel@gmail.com (P.S.); martin.buljubasich@hospitalitaliano.org.ar (M.B.); 3The Ottawa Hospital, 501 Smyth Rd., Ottawa, ON K1H 8L6, Canada; ihorton@ohri.ca (I.H.); habdelbary@toh.ca (H.A.); sigarceau@toh.ca (S.G.); 4Sheffield Teaching Hospitals, Northern General Hospital, Herries Road, Sheffield S5 7AU, UK; saifsalih@hotmail.com (S.S.); bonita.tsang28@gmail.com (B.T.); 5Auckland City Hospital, 2 Park Road, Grafton, Auckland 1023, New Zealand; paul.monk@auckland.ac.nz (P.M.); cgoplen@ualberta.ca (M.G.)

**Keywords:** periprosthetic joint infection, PJI, microbiological profile, microbiological trends, antibiotic resistance, culture

## Abstract

Periprosthetic joint infection (PJI) is a complex complication of total joint arthroplasty, with microbiological profiles varying across centers worldwide. However, most studies are limited to single-center or intra-country multicenter analyses, often including mixed cohorts of primary and revision PJI cases, with limited data regarding global antibiotic resistance patterns. This study compared the microbiological characteristics, polymicrobial culture rates, prevalence of culture-negative infections, and antibiotic resistance patterns in PJI cases across five referral centers from five continents. A total of 717 patients with primary hip and knee PJI were included from centers in Argentina, Canada, Turkey, England, and New Zealand. *Staphylococcus aureus* and *Staphylococcus epidermidis* were the most common pathogens (48.5%, *p* < 0.01). Culture-negative infection rates varied significantly, ranging from 4.2% (England) to 24.6% (Turkey) (*p* < 0.01). Polymicrobial infections were the most frequent in Canada (8.9%) and the least frequent in England (1.1%) (*p* < 0.01). Gram-negative bacteria comprised 13.1% of culture-positive cases, with no significant intercountry difference. Multidrug resistance was observed in all centers, ranging from 23.7% (Argentina) to 43.1% (Turkey), with no statistical significance. Vancomycin resistance was detected in England (2.3%) and Canada (1.2%) but absent in Turkey, New Zealand, and Argentina. These findings underscore significant intercontinental variability, emphasizing the need for regional considerations in regards to empiric antibiotic selection and PJI management.

## 1. Introduction

The incidence of periprosthetic joint infection (PJI) following total joint arthroplasty is reported to range between 0.5 and 3% [1,2,3,4]. There are established treatment strategies such as DAIR, one-stage revision, or two-stage revision, depending on the patient’s status and the chronicity of the disease [5]. One of the most important factors affecting treatment success is the isolation of the microorganism and the determination of the corresponding antibiotic sensitivity [6,7,8]. Although there are several proposed diagnostic and treatment algorithms for PJI, each center may employ different microbiological characteristics that should be taken into account [9]. For instance, a center from Brazil [10] reported a 60% incidence of monomicrobial PJIs, with *Staphylococcus aureus* being the most common offending organism, as well as 16% polymicrobial and 23% culture-negative infections, whereas a center from China [11] reported only a 8.6% incidence of polymicrobial culture results and 15.9% of culture-negative infections. A multicenter study comparing PJI referral centers from the USA and Europe concluded that the most common isolated bacteria, polymicrobial, and anaerobic infection rates differed significantly between centers [12].

Antibiotic resistance is a ubiquitous problem when treating infections that require long-term antibiotic therapy [13]. A recent study reported an incidence of antibiotic resistance of about 49.6% and 37% amongst PJI patients treated in the USA and Europe, respectively [12]. While methicillin resistance is frequent and has been well studied in PJI patients [14,15,16], vancomycin resistance is not common and is expected to be an emerging problem in PJI patients, especially for developed countries. Considering that vancomycin is one of the most common antibiotics loaded in spacers, an increase in vancomycin resistance might have serious implications, generating more difficult-to-treat infections [17]. Recently, a center from Germany [16] reported a 4% incidence of vancomycin resistance in PJI cases with *Enterococcus* species, whereas another center from China reported no vancomycin resistance [11]. Different antibiotic resistance profiles among different centers are, therefore, another factor to consider when implementing treatment algorithms to be applied universally.

The literature shows that microbiological profiles of PJI patients vary among centers from different countries [9]. However, most of the studies are either from a single center or multicenter within the same country; mixed cohorts, including primary and revision PJIs, were also used in the literature [11,18,19,20]. There is scarcity of evidence comparing antibiotic resistance profiles between multiple countries around the world. The purpose of this study was to comprehensively describe differences in the microbiological profile amongst primary hip and knee arthroplasty PJIs treated in specialist centers in five different countries/continents. In doing so, we compared the most commonly isolated microorganism, the polymicrobial culture ratio, the prevalence of culture-negative infections, and the antibiotic resistance identified.

## 2. Methods

### 2.1. Study Design and Setting

This was a retrospective, descriptive, international, multicenter cohort study conducted at five tertiary referral centers specializing in periprosthetic joint infection (PJI). The participating centers were located in Argentina (South America), England (Europe), Turkey (Asia), New Zealand (Oceania), and Canada (North America). Each center obtained approval from their institutional review boards prior to data collection.

### 2.2. Participants

We included consecutive patients aged ≥18 years who were diagnosed with periprosthetic hip or knee infection between January 2012 and January 2022, based on the 2018 Musculoskeletal Infection Society (MSIS) criteria [21].

Inclusion criterion:Diagnosis of PJI in a unilateral primary total hip or knee arthroplasty.

Exclusion criteria:
Revision arthroplasty;Bilateral arthroplasty;Arthroplasty performed for trauma or tumor reconstruction.

### 2.3. Data Collection

Data were retrospectively extracted from each hospital’s electronic medical records using a standardized template. Collected data included demographic information (age, sex), microbiological findings (most commonly isolated organisms, *Methicillin-resistant Staphylococcus aureus* (MRSA) prevalence, Gram-negative anaerobic pathogens, culture-negative cases, polymicrobial infections), and antibiotic susceptibility patterns (multidrug resistance, vancomycin resistance).

The culture results included both preoperative aspirations and intraoperative tissue samples. Culture positivity for two or more microorganisms from at least two cultures of joint aspirates or intraoperative tissue specimens or isolation in at least one intraoperative culture of two or more microorganisms, plus evidence of infection in a joint space (purulence, acute inflammation, sinus tract communicating with a joint space) was considered as polymicrobial infection [22]. Each center performed microbiological testing according to their institutional protocols. No standardized antibiotic panel was applied across all sites.

### 2.4. Outcome Measures

The primary outcomes of the study include identifying the PJI profiles at each center, presenting the most frequently isolated bacteria, determining the rates of negative and polymicrobial cultures, assessing antibiotic resistance in PJI cases, and comparing all these data across centers. The secondary outcomes comprise examining atypical culture results (e.g., anaerobic, Gram-negative bacteria), comparing multidrug resistance among centers, and investigating the emergence of vancomycin resistance.

### 2.5. Statistical Analysis

Categorical variables were summarized as frequencies and percentages. Group comparisons were conducted using Pearson’s Chi-square or Fisher’s exact test, as appropriate. A *p*-value < 0.05 was considered statistically significant. For significant results in Chi-square tests, post hoc pairwise comparisons were performed using Bonferroni correction. The pairwise comparisons between the countries are presented in tables. In case there is a significant difference between pairs, the exact *p* value is given under the larger proportion. All analyses were performed using the Statistical Package for Social Sciences software (SPSS) 23.0 (SPSS Inc., Chicago, IL, USA).

## 3. Results

A total of 717 patients (338 hips; 379 knees) were included in the study, with 302 patients (143 hips; 159 knees) from Canada, 190 patients (82 hips; 108 knees) from England, 97 patients (63 hips; 34 knees) from Argentina, 65 patients (20 hips; 45 knees) from Turkey, and 63 patients (30 hips; 33 knees) from New Zealand. The mean age of the whole cohort was 68.2, with ages ranging from 33 to 92. A total of 52% (373/717) of the cohort was female.

*Staphylococcus aureus* (23.2%) and *Staphylococcus epidermidis* (17.0%) were the most commonly isolated microorganisms for the whole cohort (*p* < 0.01). They constituted 48.5% of the culture-positive cases. The most commonly isolated microorganisms were *Staphylococcus epidermidis* in England and Turkey, and *Staphylococcus aureus* in Canada, Argentina, and New Zealand (Table 1). A total of 55.2% of the polymicrobial PJI cases included either *Staphylococcus aureus* or *Staphylococcus epidermidis.* With the addition of *Staphylococcus aureus* and *Staphylococcus epidermidis* from the polymicrobial PJI cases, these pathogens were identified in up to 53.7% of culture-positive cases (*p* < 0.01).

Culture-negative PJIs had an overall prevalence of 17.3% in all cohorts, ranging between 4.2% (England) to 23.5% (Canada), with a statistically significant between-country difference (*p* < 0.01). The overall ratio of polymicrobial PJI was 5.6% among all cases. It ranged from 1.1% (England) to 8.9% (Canada) between countries (*p* < 0.01) (Figure 1).

The prevalence of Gram-negative bacteria among culture-positive cases was found to be 13.1% in all cohorts. It ranged from 7.4% (New Zealand) to 20.0% (Turkey) (*p* > 0.05). The prevalence of anaerobic bacteria was found to range between 0.0% (Turkey) and 7.5% (Canada), with a mean of 5.3% (*p* > 0.05). The prevalence of MRSA was 5.1% among culture-positive cases, ranging from 1.8% (New Zealand) to 16.0%(Argentina) (*p* < 0.001).

A total of 58% of the microorganisms, ranging from 38.1% (New Zealand) to 62.9% (Argentina), showed complete antibiotic susceptibility in the study (*p* = 0.02). Multidrug resistance was observed in 36.7% of the cohort, ranging between 23.7% (Argentina) to 43.1% (Turkey), with no statistically significant difference among countries. The rest of the cohort showed resistance to only one antibiotic (Figure 2). Vancomycin resistance was reported in England (hip 1.2, knee 2.7%) and Canada (hip 0.7, knee 1.2%); vancomycin resistance was not seen (0%) in Turkey, New Zealand, or Argentina (*p* > 0.05). No predisposition to antibiotic resistance was observed when comparing microorganisms based on Gram-staining characteristics (Gram-positive vs. Gram-negative), metabolism (aerobic vs. anaerobic), or joint type (hip vs. knee).

Pairwise comparisons of each country regarding culture results and antibiotic resistance are presented in Table 2 and Table 3. When hip and knee patients were compared with each other for the whole study group, no difference was observed in terms of negative-culture results, single- or multi-antibiotic resistance, Gram-negative bacteria prevalence, or MRSA prevalence (*p* > 0.05). Hip PJIs displayed a greater prevalence of polymicrobial culture rates compared to that of knee PJIs (7.4% vs. 4.0%, *p* = 0.02). Also, hip PJIs showed a greater prevalence of anaerobic bacteria compared to that of knee PJIs (7.8% vs. 3.1%, *p* = 0.01).

## 4. Discussion

PJI remains a significant challenge in orthopedic surgery, with considerable variability in microbiological profiles observed across different regions worldwide. This multicenter international study aimed to elucidate these differences, providing insights into the culture results and antibiotic resistance patterns among PJI patients from five centers located on different continents. This issue is important when elaborating universal guidelines and treatment algorithms generated from consensus meetings [23].

The main findings of this study underscored the diversity of microbiological profiles among the participating centers. *Staphylococcus* species, particularly *Staphylococcus aureus* and *Staphylococcus epidermidis*, emerged as the predominant pathogens across all centers, consistent with the results of previous literature [24,25,26] highlighting their prominent role in PJI (Table 4). Notably, the distribution of these organisms varied among countries, with differences observed in the most commonly isolated microorganism between hip and knee infections. While *Staphylococcus aureus* predominated in Canada, Argentina, and New Zealand, *Staphylococcus epidermidis* was more prevalent in England and Turkey. Similar observations have been reported previously by others; Casenaz et al. described the culture results of 282 PJI cases from France, reporting that *Staphylococcus aureus* was the most commonly isolated pathogen, with a prevalence of 44.3%, followed by coagulase-negative *Staphylococci*, with a ratio of 25.2% [14]. Villa et al. compared the microbiological profiles of PJI cases in seven centers from six countries [9]. In this study, *Staphylococcus aureus* was the most frequently isolated microorganism in the USA, England, Uruguay and Russia, with ratio of between 21.4 and 34.6%, whereas *Staphylococcus epidermidis* was most commonly detected in Argentina and Germany, with a ratio of between 25.2–27.1% [9]. The microorganisms identified beyond the most commonly observed types were presented in a detailed list to serve as a reference. Although there were notable variations in the distribution of less frequently detected microorganisms across different centers, the low sample sizes precluded the identification of statistically significant differences. To more accurately delineate international variations, studies with larger cohorts are required.

Polymicrobial infections, although less frequent, were still notable in the current study, ranging from 1.1 to 8.9%. Canada displayed the highest rate of polymicrobial cultures, which is consistent with the results in the literature. Kandel reported 9% polymicrobial culture results in 533 PJI cases from a Canadian referral center [28]. Polymicrobial infections are often associated with wound complications/sinus tracts, the presence of treatment-resistant organisms in the flora, and patients with high comorbidity [27]. Additionally, the sebaceous gland-rich nature of the groin region and its proximity to the anogenital tract are thought to contribute to the higher prevalence of anaerobic and polymicrobial infections following total hip arthroplasty compared to that for total knee arthroplasty, which was the case for this study [29]. In the current study, the frequent use of the anterior approach to hip replacement in the Canadian center may have also played a role in the occurrence of polymicrobial infections. The presence of polymicrobial flora poses additional challenges in treatment decision making, as it may necessitate broader-spectrum antimicrobial coverage and various surgical interventions [30]. Furthermore, the treatment success in polymicrobial PJI cases is lower compared to that in monomicrobial cases. Tan et al. reported that patients with polymicrobial PJI had a higher treatment failure rate (50.5%) and were more likely to undergo arthrodesis (OR = 11.06) or amputation (OR = 3.80) and had an increased risk of mortality (OR = 7.88) compared to the results for those with monomicrobial PJI [31].

Another notable finding of the current study was the considerable proportion of negative-culture results, ranging from 4.2 to 24.6% across centers. The literature supports the variability in negative-culture rates, which can be observed at diverse rates across different countries and even between different centers within the same country. Runner et al. compared PJI trends of two hospitals from the USA [19]. The results revealed significant differences in negative-culture rates between centers (4.6 vs. 15.8%). Culture-negative PJIs present several challenges. The lack of targeted antibiotic therapy can negatively impact treatment success. Tan et al. reported a 69.2% treatment success for culture-negative PJI cases [32]. Infections with unidentified pathogens often require broad-spectrum antibiotics, increasing the risk of drug interactions and systemic toxicity for the patient, while also contributing to global antibiotic resistance [33]. Additionally, attempting to treat an infection without a known etiology may cause anxiety and psychological distress in some patients [33]. These findings underscore the need for improved diagnostic techniques to enhance pathogen detection rates. There are fundamental principles which should be followed in order to enhance culture yield. Samples should be collected from infected areas, with an incubation period of at least 5 days, extending to 14 days for slow-growing organisms [34]. Specialized media should be used for certain bacteria [35]. It is recommended to obtain at least 3, and ideally, 5–6, samples for optimal diagnostic accuracy [36]. To grow biofilm-forming microorganisms, techniques such as sonication or chemical disruption with dithiothreitol (DTT) can be used to break down the biofilm layer [37]. Molecular diagnostic techniques, such as multiplex PCR, 16S rRNA sequencing, and next-generation sequencing (NGS), have significantly improved the identification of pathogens in periprosthetic joint infections (PJIs), overcoming the limitations of traditional cultures, particularly in culture-negative cases [33]. NGS-based methods, including metagenomic and metatranscriptomic sequencing, can detect a wide range of organisms (including nonviable and antibiotic-treated bacteria). In a study by Goswami et al. on 85 patients with culture-negative PJI, NGS was used to identify the causative organisms. They demonstrated that bacteria could be identified in 65.9% of culture-negative PJI cases using NGS [38]. Moreover, NGS-based methods may provide insight into antimicrobial resistance and active infections. However, these insights are primarily based on the presence of known resistance genes and mutations rather than direct phenotypic susceptibility testing. There are also instances in which genetic detection did not correlate with actual resistance [39].

The antibiotic resistance patterns revealed concerning trends, with notable variations observed between countries. Resistance to at least one antibiotic was observed in 37.1 to 61.9% of PJI cultures in the study group. These findings are consistent with those in the previous literature. Villa et al. reported a 37.7 (USA) to 77.9% (Russia) resistance to at least one antibiotic [9]. In the current study, multidrug resistance was the highest in Turkey, with a rate of 43.1%, and the lowest in Argentina and New Zealand, with a rate of 23.7 and 23.8%, respectively. Multiple factors, such as the existence of antibiotic stewardship programs, healthcare practices, infection control measures, antibiotic use in agriculture, public awareness, and access to antibiotics may account for significant difference in antibiotic resistance rates. In their meta-analysis including 11 countries, Rabbi et al. reported a significantly higher rate of antibiotic over-prescription in high-income countries compared to that for middle- or low-income countries [40]. Aggarwal et al. identified several reasons for differences in methicillin resistance between reference centers in Europe and the United States [12]. The study found that the average BMI was higher in the US center, which could contribute to increased antibiotic resistance due to inadequate antibiotic dosing in obese patients. Additionally, differences in clinical practices may have influenced resistance rates. For example, vancomycin use in the European center was restricted to MRSA cases, whereas in the US center, it was used more broadly. Variations in infection control policies between the centers were also highlighted as a contributing factor. Access to antibiotics is another factor contributing to resistance. The rate of the dispensing of unprescribed antibiotics from community pharmacies was the lowest in the high-income countries (48.4%), and the highest in the low-income countries (75.1%) [41]. Turkey was reported to have the highest rate of antibiotic use among 12 Eastern European countries [42]. In a study comparing antibiotic use for humans and food animals among 26 European countries, Australia, Canada, New Zealand, and the United States of America, New Zealand was found to be the third lowest user of antimicrobials in animal production and the sixteenth in humans [43]. These findings could explain the different multidrug resistance rates in the current study.

Of particular concern was the emergence of vancomycin resistance, which poses a significant threat to treatment options, as vancomycin is commonly used in spacers during two-stage revisions and in systemic therapy regimes [7]. Numerous studies have documented a high prevalence of methicillin resistance in PJI cases [11,15,16,44]. Consequently, vancomycin has become extensively utilized due to its broad coverage. In some centers, vancomycin is even employed prophylactically in primary cases [45,46,47]. However, the routine use of vancomycin is likely to foster the development of increasingly resistant strains over time [48]. Indeed, centers from various countries have reported instances of vancomycin-resistant PJI cases. Shabana et al reported decreased susceptibility to vancomycin with an MIC ≥ 2 mg/L in 70% of PJI patients with *S. Epidermidis* [44]. Drago et al. reported a 1.7% vancomycin resistance in knee PJI patients with a culture-positivity for *S. Epidermidis* [24]. Aggarwal et al. reported a 26.7% vancomycin resistance in 30 PJI cases with *Enterococcus* in the center from the USA, whereas there was no resistance in the center from Germany [12]. In the current study, the center from England reported a 1.2 and 2.7% vancomycin resistance, and the center from Canada reported a 0.7 and 1.2% vancomycin resistance in hip and knee PJI cases, respectively. Although the overall prevalence of vancomycin resistance is generally low, its presence underscores the importance of surveillance and infection control measures to prevent its dissemination.

The limitations of our study include its retrospective nature and the inherent biases associated with multicenter analyses. Additionally, differences in diagnostic criteria and laboratory methodologies—including culture duration, techniques, and medium—among centers may have influenced the reported microbiological profiles and resistance patterns. Other than vancomycin, antibiotic resistance was not specified in the current study. Although the study provides insights about antibiotic resistance profiles, a detailed list specifying the antibiotics for which resistance was observed could be valuable in future studies. The culture results of the patients included in the study reflect those obtained through conventional methods. More advanced technologies, such as NGS, can yield more accurate and higher success rates for culture results [49,50]. Incorporating these techniques could potentially reduce the high culture-negative rates observed in our study. This study confirms the hypothesis that there were differences among centers in terms of the microbiological profile in PJI cases. It is important to acknowledge that while the five institutions included in this study are representative of the patient populations in their respective geographical regions, the findings should be interpreted cautiously, as they may not fully reflect all potential centers within the corresponding countries. Future prospective studies with standardized protocols are warranted to validate our findings and further elucidate the factors contributing to regional variations in PJI microbiology and antibiotic resistance.

In conclusion, our international multicenter study provides valuable insights into the microbiological profiles and antibiotic resistance patterns of PJI across different regions worldwide. The observed variability underscores the need for tailored approaches to diagnosis and treatment, taking into account local epidemiology and resistance patterns. Notably, the culture-negativity rates observed at some centers highlight the challenge of guiding antimicrobial therapy in the absence of an identified pathogen. These findings suggest that understanding regional resistance patterns and common microbiological trends can help inform empirical antibiotic choices in culture-negative cases, optimizing post-operative management. Collaborative efforts remain essential to address the challenges posed by PJI and mitigate the impact of antimicrobial resistance on patient outcomes.

## Figures and Tables

**Figure 1 microorganisms-13-01505-f001:**
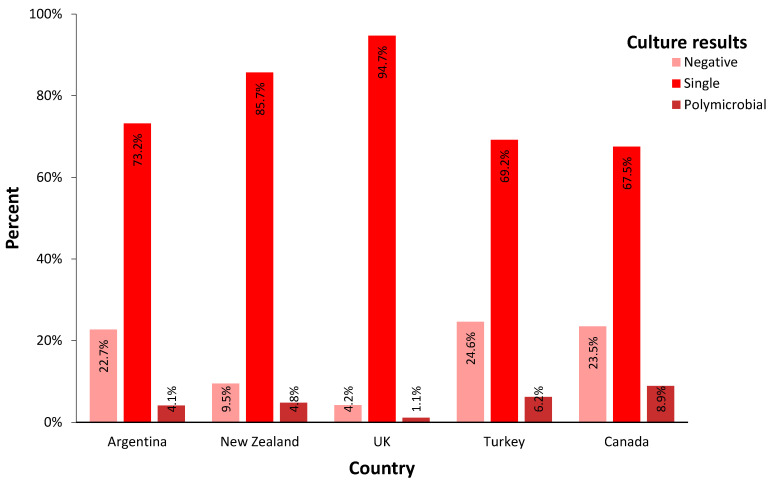
Incidence of culture results from PJI patients in each country. (single: culture results showed growth of one bacterium; polymicrobial: culture results showed growth of two or more bacteria).

**Figure 2 microorganisms-13-01505-f002:**
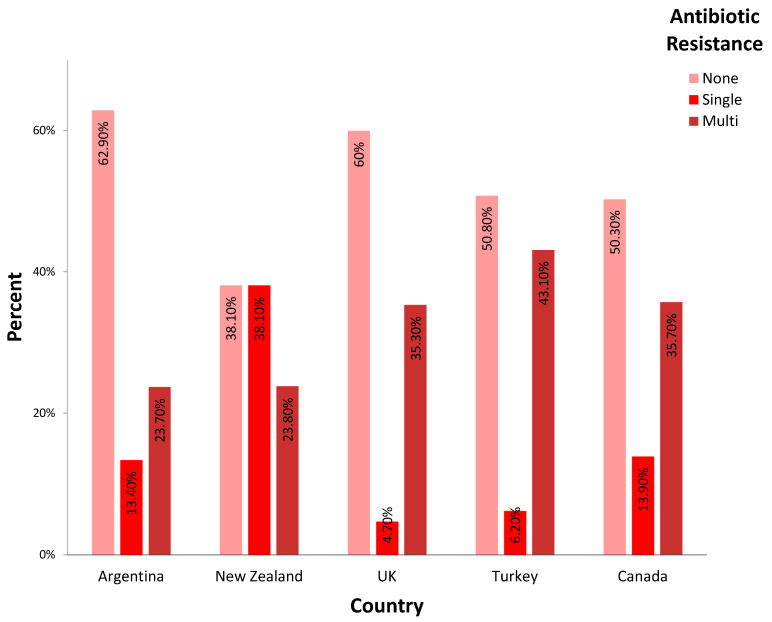
Incidence of antibiotic resistance of the diverse offending organisms in each country. (None: no antibiotic resistance; single: resistance to only one antibiotic; multi: resistance to two or more antibiotics was reported).

**Table 1 microorganisms-13-01505-t001:** List of microorganisms identified among the five centers. *Staphylococcus aureus* and *Coagulase*-negative *staphylococci* were the most common bacteria, with significant differences among centers. *Cutibacterium acnes* and *E. coli* were the most common anaerobic and Gram-negative bacteria among whole cohort, respectively, with no significant difference among centers. (Note: Proportions were calculated among culture-positive cases; culture-negative cases were excluded.)

Microorganism	Argentina n (%)	New Zealand n (%)	England n (%)	Turkey n (%)	Canada n (%)	*p* Value
Aerobic gram-positive						
** *Staphylococcus aureus* **	31 (41.3)	23 (40.4)	31 (17)	12 (24.5)	69 (30)	**<0.001**
***Coagulase*-negative**	13 (17.3)	7 (12.3)	38 (20.9)	18 (36.7)	46 (20)	**0.03**
** *Staphylococci* **						
*Streptococcus* species	11 (14.4)	14 (24.6)	15 (8.2)	4 (8.1)	34 (14.8)	>0.05
Other *Staphylococcus* sp.	5 (6.5)	5 (8.7)	17 (9.3)	1 (2)	4 (1.7)	>0.05
*Listeria monocytogenes*	1 (1.3)	.	.	.	.	>0.05
*Enterococcus faecalis*	.	1 (1.7)	17 (9.3)	.	13 (5.6)	>0.05
*Enterococcus faecium*	.	.	2 (1)	.	.	>0.05
*Corynebacterium* species	.	.	8 (4.4)	1 (2)	.	>0.05
*Bacillus* sp.	.	.	4 (2.2)	.	.	>0.05
Unspecified	.	.	3 (1.6)	.	.	>0.05
Anaerobes						
** *Cutibacterium acnes* **	1 (1.3)	.	4 (2.2)	.	6 (2.6)	>0.05
*Peptostreptococcus* species	.	.	3 (1.6)	.	.	>0.05
*Clostridium* species	.	.	3 (1.6)	.	1 (0.4)	>0.05
*Bacteroides fragilis*	.	.	2 (1)	.	.	>0.05
*Prevotella species*	.	.	.	.	.	>0.05
*Finegoldia magna*	.	1 (1.7)	2 (1)	.	5 (2.2)	>0.05
Unspecified	.	.	.	.	1 (0.4)	>0.05
Gram-negative						
** *E. coli* **	3 (3.9)	2 (3.5)	7 (3.8)	5 (10.2)	11 (4.8)	>0.05
*Klebsiella* sp.	1 (1.3)	.	5 (2.7)	2 (4)	2 (0.8)	>0.05
*Enterobacter* species	1 (1.3)	.	2 (1)	.	1 (0.4)	>0.05
*Proteus mirabilis*	.	.	4 (2.2)	.	1 (0.4)	>0.05
*Salmonella enterica*	.	.	.	.	1 (0.4)	>0.05
*Pseudomonas aeruginosa*	2 (2.6)	.	2 (1)	1 (2)	2 (0.8)	>0.05
*Morganella morganii*	2 (2.6)	.	1 (0.5)	.	2 (0.8)	>0.05
*Burkholderia cepacian*	.	.	.	1 (2)	.	>0.05
*Serratia marcesens*	.	1 (1.7)	1 (0.5)	.	1 (0.4)	>0.05
*Gardnerella adiacens*	.	.	3 (1.6)	.	1 (0.4)	>0.05
Unspecified	.	.	5 (2.7)	.	.	>0.05
Fungus						
*Candida* species	1 (1.3)	.	1 (0.5)	.	2 (0.8)	>0.05
**Polymicrobial**	4 (5.2)	3 (5.3)	2 (1)	4 (8.1)	27 (11.7)	**0.003**
**Total #**	76	57	182	49	230	

**Table 2 microorganisms-13-01505-t002:** Pairwise comparison of culture results from each country. (Note: for each significant pair, *p* value ( ) is given under the larger proportion.) Note that England displayed a significantly lower rate of negative-cultures than did Canada, Turkey, and Argentina. Canada showed a significantly higher rate of polymicrobial culture results than did England.

Comparison of Culture Results Among Countries								
			Country					
			Turkey (TR)	Argentina (AR)	New Zealand (NZ)	England (UK)	Canada (CA)	Total
All patients	Culture results	Negative %	24.6 **UK (<0.001)**	22.7 **UK (<0.001)**	9.5	4.2	23.5 **UK (<0.001)**	17.2
		Single %	69.2	73.2	85.7 **CA (0.04)**	94.7 **TR (<0.001)** **AR (< 0.001)** **CA (< 0.001)**	67.5	77.3
		Poly %	6.2	4.1	4.8	1.1	8.9 **UK (0.003)**	5.6
	Total	case #	65	97	63	190	302	717
Hip	Culture results	Negative %	30.0 **UK (0.02)**	27.0 **UK (0.005)**	6.7	6.1	18.6	16.3
		Single %	65.0	68.3	86.7	92.7 **TR (0.009)** **AR (0.001)** **CA (0.001)**	69.0	76.3
		Poly %	5.0	4.8	6.7	1.2	12.3 **UK (0.03)**	7.4
	Total	case #	20	63	30	82	143	338
Knee	Culture results	Negative %	22.2 **UK (0.001)**	14.7	12.1	2.8	28.9 **UK (<0.001)**	17.9
		Single %	71.1	82.4	84.8	96.3 **TR (<0.001)** **CA (<0.001)**	65.4	78.1
		Poly %	6.7	2.9	3.0	0.9	5.7	4.0
	Total	case #	45	34	33	108	162	379

**Table 3 microorganisms-13-01505-t003:** Pairwise comparison of antibiotic resistance from each country. (Note: for each significant pair, *p* value ( ) is given under the larger proportion.) Note that New Zealand displayed a significantly greater antibiotic susceptibility than did Argentina and England. Multidrug resistance was prevalent among all centers, with no significant difference. Vancomycin resistance was seen only in England and Canada.

Comparison of Antibiotic Resistance Among Countries								
			Country					
			Turkey (TR)	Argentina (AR)	New Zealand (NZ)	England (UK)	Canada (CA)	Total
All patients	Antibiotic resistance	None %	50.8	62.9 **NZ (0.02)**	38.1	60.0 **NZ (0.02)**	50.3	53.6
		Single %	6.2	13.4	38.1 **TR (<0.001)** **AR (0.003)** **UK (<0.001)** **CA (<0.001)**	4.7	13.9 **UK (0.01)**	12.8
		Multi %	43.1	23.7	23.8	35.3	35.8	33.6
	Total	case #	65	97	63	190	302	717
		Vancomycin Resistance %	0.0	0.0	0.0	2.1	1.0	0.9
Hip	Antibiotic resistance	None %	55.0	71.4 **NZ (0.01)** **CA (0.03)**	36.7	57.3	49.7	54.7
		Single %	5.0	6.3	30.0 **AR (0.02)** **UK (0.002)** **CA (0.03)**	4.9	9.8	9.5
		Multi %	40.0	22.2	33.3	37.8	40.6	35.8
	Total	case #	20	63	30	82	143	338
Knee	Antibiotic resistance	None %	48.9	47.1	39.4	62.0	50.9	52.5
		Single %	6.7	26.5 **UK (.02)**	45.5 **TR (0.001)** **UK (<0.001)** **CA (0.005)**	4.6	17.6 **UK (0.01)**	15.8
		Multi %	44.4	26.5	15.2	33.3	31.4	31.7
	Total	case #	45	34	33	108	159	379

**Table 4 microorganisms-13-01505-t004:** Literature review of multicenter studies reporting microbiological profiles in hip and knee PJI cases.

Study	# Case	Countries	The Most Common Organism (Prevalence)	Negative-Culture Rate	Poly-Microbial Culture Results	Gram neg. Bacteria	Anaerobic Bacteria	Antibiotic Susceptibility	MRSA Prevalence	Point of Interest
**Villa et. al., 2021** [9]	654	All cohorts	*Staphylococcus aureus* (24.8%)	n/a	9.3%	n/a	n/a	42%	n/a	The rates of resistant organisms and polymicrobial infections vary significantly among countries.
		USA	*Staphylococcus aureus* (21.4%)		9.4%			62.3%		
		UK	*Staphylococcus aureus* (21.4%)		4.9%			59.2%		
		Uruguay	*Staphylococcus aureus* (34.6%)		4.6%			28.5%		
		Argentina	*Staphylococcus epidermidis* (25.2%)		11.1%			33.3%		
		Germany	*Staphylococcus epidermidis* (27.1%)		11.9%			37.3%		
		Russia	*Staphylococcus aureus* (29.8%)		16.3%			22.1%		
**Aggarwal et. al., 2014** [12]	1670	USA	*Staphylococcus aureus* (31.0%)	15.8%	7.4%	6.6%	0.9%	n/a	48.1%	The infecting organisms in PJI vary between an orthopedic center in Europe and one in the USA, with higher virulence and greater antibiotic resistance observed in the American institution.
		Germany	*Staphylococcus epidermidis* (39.3%)	16.1%	3.4%	4.3%	9%		12.8%	
**Peng et. al., 2021** [27]	925	34 centers within China	*Staphylococcus aureus* (26.5%)	9.8%	26.9%	8.2%	3.5%	n/a	10.5%	PJI-causing organisms differed between hip and knee joints, with enteric Gram-negative bacilli, anaerobes, and polymicrobial infections being more common in prosthetic hip joints.

## Data Availability

The datasets presented in this article are not readily available. Due to the international multicenter nature of this study, each center follows its own data-sharing protocol, and a unified data-sharing protocol could not be established. Requests to access the datasets should be directed to ggrammatopoulos@toh.ca.

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
