# Peer review of "Comparison of Microbiological Profiles of Primary Hip and Knee Peri-Prosthetic Joint Infections Treated at Specialist Centers Around the World"

_microorganisms, 2025, doi:10.3390/microorganisms13071505_

Round 1
Reviewer 1 Report
Comments and Suggestions for Authors
Thank you for the opportunity to review this manuscript. The paper is well written, with clear presentation of results and a solid structure throughout. The data presented are highly relevant and offer valuable insights for tailoring prophylaxis, prevention, and treatment protocols in periprosthetic joint infections.
In my opinion, this is an excellent study and fully deserves approval. It would be interesting, in future research, to investigate whether the observed microbiological differences among centers may correlate with different outcomes in DAIR procedures—while acknowledging the numerous potential biases such an analysis would entail.
Congratulations to the authors for this outstanding work.
Author Response
Dear Madam/Sir,
Thank you very much for taking the time to review this manuscript. We are very pleased to
get such kind comments. Your future research suggestion regarding DAIR is very inspiring
and definitely would be a topic of high quality research. Thank you once again for your
valuable insights, and we hope to cross paths again in future projects.
King regards,
Authors
Reviewer 2 Report
Comments and Suggestions for Authors
The paper is interesting, grammatically well written, and highlights many aspects of the PJI, which is a strong topic. However, there are a few comments that I would like authors to address.
-First, it's hard to understand when the authors describe a figure. I would suggest that for each figure, the authors will quickly introduce it and then focus on the specific.
-The authors mention several bacteria listed in Table 1. I would also suggest plotting the main ones that have a significant influence, like E. coli and others, to highlight the differences between countries.
-About the antibiotic residence, no table or detailed info about a specific type of antibiotic used, which bacteria, etc., could you please provide this info clearly and more specifically?
-Authors describe the vancomycin resistance, but no data are easily reported or described. I suggest that the authors reorganize the information and report more detailed tables and graphs. This could also help in the discussion section to compare or explain your observation and validate the results.
Author Response
Please see attached file for full response
